# Humoral and T-Cell Response before and after a Fourth BNT162b2 Vaccine Dose in Adults ≥60 Years

**DOI:** 10.3390/jcm11092649

**Published:** 2022-05-08

**Authors:** Erez Bar-Haim, Noa Eliakim-Raz, Amos Stemmer, Hila Cohen, Uri Elia, Asaf Ness, Muhammad Awwad, Nassem Ghantous, Neta Moskovits, Shahar Rotem, Salomon M. Stemmer

**Affiliations:** 1Department of Biochemistry and Molecular Genetics, Israel Institute for Biological Research, Ness Ziona 7410001, Israel; hilac@iibr.gov.il (H.C.); urie@iibr.gov.il (U.E.); shaharr@iibr.gov.il (S.R.); 2Department of Medicine E, Rabin Medical Center, Beilinson Hospital, Petah Tikva 4941492, Israel; noaer@clalit.org.il (N.E.-R.); asaf.ness@gmail.com (A.N.); m.awwad15@outlook.com (M.A.); naseem2410@gmail.com (N.G.); 3Infectious Diseases Unit, Rabin Medical Center, Beilinson Hospital, Petah Tikva 4941492, Israel; 4Sackler Faculty of Medicine, Tel Aviv University, Tel Aviv 6997801, Israel; salomon.stemmer@gmail.com or; 5Department of Oncology, Sheba Medical Center, Tel Hashomer, Ramat Gan 5224213, Israel; amos.ste@gmail.com; 6Felsenstein Medical Research Center, Sackler Faculty of Medicine, Tel Aviv University, Tel Aviv 6997801, Israel; neta.moskovits@gmail.com; 7Davidoff Center, Rabin Medical Center, Beilinson Hospital, Petah Tikva 4941492, Israel

**Keywords:** SARS-CoV-2, BNT162b2, COVID-19

## Abstract

Both humoral and cellular anamnestic responses are significant for protective immunity against SARS-CoV-2. In the current study, the responses in elderly people before and after a fourth vaccine dose of BNT162b2 were compared to those of individuals immunized with three vaccine doses. Although a boost effect was observed, the high response following the third administration questions the necessity of an early fourth boost.

## 1. Introduction

The SARS-CoV-2 B.1.1.529 variant (Omicron) wave in Israel led to the early authorization of a fourth dose of the BNT162b2 vaccine (BioNTech/Pfizer) to individuals with age ≥60 years who had received a third dose at least 4 months earlier. The potential benefit of a third vaccine dose of BNT162b2 or mRNA-1273 was demonstrated by lower rates of breakthrough infection and effectiveness against emerging variants of concern (VOCs) [1]. Although currently, there are no clear correlates of protection, the involvement of both humoral and T-cell immunity in protection from COVID-19 was shown [2]. Age-associated immune system dysfunction, manifested by compromised immunity parameters such as declined lymphocyte function may eventually predispose one to severe COVID-19. Consequently, the vaccination of such individuals might be beneficial [3].

We characterized the humoral and cellular immune responses prior and following a fourth BNT162b2 vaccine dose and compared them to the responses amongst individuals four months following a third vaccine dose.

## 2. Materials and Methods

Participants ≥60 years (*n* = 16) without prior SARS-CoV-2 infection or active malignancy were recruited in the Rabin Medical Center (RMC) vaccination center. The study was approved by the ethics committee of RMC, and all participants provided written informed consent.

Anti-spike IgG titers and T-cell response against the ancestral and Omicron spike proteins were determined as previously described [4,5]. Anti-S IgG titers were determined in the serum with the SARS-CoV-2 IgG II Quant assay (Abbott Laboratories, Lake Forest, IL, USA) with strict adherence to the manufacturer’s protocol. Seropositivity was defined as ≥50 arbitrary units (AU)/mL.

For T-cell response, blood was collected into sodium-heparin tubes (vacutainer, BD, Franklin Lakes, NJ, USA) and processed within 2 h of collection. Peripheral blood mononuclear cells (PBMCs) were isolated with density gradient sedimentation using Ficoll-Paque (Sigma-Aldrich, Rehovot, Israel) according to the manufacturer’s protocol. PBMCs were stimulated with commercially available peptide pools (15-mer sequences with an overlap of 11 amino acids) covering the full length of the Wuhan-1 SARS-CoV-2 (wild-type) or Omicron B.1.1.529 variant spike (Peptides & Elephants GmbH, Hennigsdorf, Germany). Interforon gamma (IFNɣ)-secreting cells were quantified using a fluorescent ELISPOT assay (ImmunoSpot, Cleveland, OH, USA) with strict adherence to the manufacturer’s protocol. Data were acquired with the ImmunoSpot S6 Ultimate reader and analyzed with ImmunoSpot software version 7.0.30.2 (ImmunoSpot). A positive T-cell response was defined as ≥10 IFNɣ-secreting cells per 10^6^ PBMCs. The presented T response is the average of four measurements minus background response without antigen stimulation. Samples with background responses ≥25 spots were excluded (not applicable, NA). 

## 3. Results and Discussion

All 16 participants in the study were evaluated 20 (T1) and 22 (T2) weeks after the third dose. Among the 16 participants, 5 participants received a fourth dose at week 20 (after the blood draw); 9 received three doses only, and 2 who received only three doses had a polymerase chain reaction (PCR)-confirmed SARS-CoV-2 infection between T1 and T2.

In the five participants with four doses, who were all seropositive before the fourth dose (T1), the anti-spike IgG levels increased (4.0–11.3-fold) after the fourth dose (T2). At T1, four and two of the five participants had a T-cell response to the ancestral and Omicron spike protein, respectively. At T2, all five had a T-cell response against both spike proteins that was generally higher than before the fourth dose (Table 1).

All nine participants with three vaccine doses were seropositive at both timepoints, although a decrease in anti-spike IgG levels was noted from T1 to T2 (1.1–1.3-fold). In T2, of the nine participants with three vaccine doses, eight had a T-cell response against the ancestral spike protein and eight had a response against the Omicron protein.

The two participants with a documented SARS-CoV-2infection demonstrated an increase in anti-spike IgG titers following the infection. For one of these participants, data on T-cell response before and after the infection were available, and an increased T-cell response against both the ancestral and Omicron spike proteins was noted.

Among all 16 participants, the average response to the ancestral spike protein was similar to that against the Omicron spike protein (average [SD] of 261.4 [401.5] vs. 80.7 [100.4]).

Sample 1 of the PCR-confirmed SARS-CoV-2 denoted an opportunity to follow the boost response following infection, most probably by the Omicron variant. Interestingly, the boost response to the Omicron spike was significantly higher than to the ancestral spike (14.5 vs. 2.8-fold increase, respectively), possibly a result of novel T-cell epitopes in the Omicron variant.

Data on the efficacy of the fourth dose are limited, and our study is the first to examine the immune response following a fourth BNT162b2 vaccine dose. The available data suggest that the fourth dose lowers the risk of infection and severe disease by 2- and 4-fold, respectively, compared to three doses [6]. In another study, a limited protective effect of the fourth vaccine against Omicron was described, in parallel to immunological boost [7]. Our study, although limited by the small sample size, provides immunogenicity data demonstrating that the majority of participants had a detectable T-cell response 20–22 weeks after the third dose regardless of the fourth dose and that the T-cell response against the Omicron spike protein was comparable to that against the ancestral spike protein. T-cell response varies between individuals due to HLA polymorphism. Additionally, it was shown that along the spike protein, for each individual, there is a median of 11 and 10 recognized epitopes of CD4 and CD8 T-cell populations, respectively [8]. Therefore, it could be speculated that T-cell response may be maintained against VOCs [5,8].

Taken together, our data show a significant humoral and cellular immune response among elderly individuals 20 weeks after a third BNT162b2 vaccine dose. Thus, given the low decay kinetics of memory B and T cells [9], our findings, as those of other studies do [7], question the benefit of an early boost.

## Figures and Tables

**Table 1 jcm-11-02649-t001:** Anti-spike IgG titers and T-cell response in participants who received 3 or 4 BNT162b2 vaccine doses.

#	Age	Sex	Anti-Spike IgG, AU/mL		T-Cell Response, IFNɣ Secreting Cells Per 10^6^ PBMC			
					Ancestral		Omicron	
			T1 (20 wks after Dose 3)	T2 (22 wks after Dose 3)	T1 (20 wks after Dose 3)	T2 (22 wks after Dose 3)	T1 (20 wks after Dose 3)	T2 (22 wks after Dose 3)
**Participants with 4 vaccine doses (4th dose was on T1, immediately after blood draw)**								
1	66	F	11,295	80,000	205	1823	117	905
2	68	F	4906	26,951	75	151	56	136
3	72	F	3696	19,711	53	372	0	219
4	65	F	2971	33,420	3	102	0	34
5	69	M	897	3547	22	29	8	11
**Participants with 3 vaccine doses**								
1	73	M	12,033	10,816	148	7	100	17
2	74	F	9113	7882	NA	78	NA	55
3	64	F	6980	6473	180	199	149	144
4	64	F	6230	5574	7	31	0	20
5	68	F	5519	4980	NA	10	NA	7
6	77	M	4476	3597	196	86	106	48
7	76	F	4009	3238	1450	1577	68	144
8	71	M	3432	2641	358	88	414	80
9	75	F	2328	2016	NA	1030	NA	108
**Participants with 3 vaccine doses and confirmed COVID-19 infection (by PCR) between T1 and T2**								
1	77	M	1348	7883	243	691	27	393
2	72	M	587	1050	NA	100	NA	56

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
