# Peer review of "Humoral and T-Cell Response before and after a Fourth BNT162b2 Vaccine Dose in Adults ≥60 Years"

_jcm, 2022, doi:10.3390/jcm11092649_

Round 1
Reviewer 1 Report
Dear authors,
The findings reported in the manuscript are impressive and clearly raise concerns regarding the necessity of a 4th mRNA vaccine dose shortly after the 3rd dose.
Although I do not have specific comments on such a complete study, some minor revisions are required.
Therefore, I suggest the following:
- The study titled "4th Dose COVID mRNA Vaccines' Immunogenicity & Efficacy Against Omicron VOC" (medRxiv, 2022, doi: https://doi.org/10.1101/2022.02.15.22270948) should be mentioned in the manuscript. Regev-Yochay G, et al, reported the safety, the immunogenicity and the efficacy of the 4th dose of mRNA COVID-19 vaccines in health care workers (HCW), whereas data regarding the cellular immune response wer also preseneted. Despite the fact that the elderly and the vulnerable populations were not assessed, the 4th vaccination seemed to have marginal benefits on HCW.
- In the sentences 64-66 ("Five participants received a 4th dose ... had a PCR-confirmed COVID-19 infection between T1 and T2"), there is a slight confusion regarding data presentation and the reader might be misled. Consequently, it should be written more clearly.
- References: i) The study led by Bar-On YM, et al, was recently publised in NEJM and the reference no 6 should be corrected. ii) DOI number can be removed from all the references.
Reviewer 2 Report
In a small cohort (n=16) participants, authors investigated the B and T-cell responses to 3rd (n=11) and 4th (n=5) doses of BNT162b2 mRNA vaccine. For B-cell responses, anti-Spike IgG levels were measured post-vaccination (but not neutralizing antibody levels), and for T-cell responses, IFN-gamma secreting cells were measured post-vaccination – both at two time points 20- and 22-weeks post-dose 3, and 0- and 14-days post-dose 4; in addition, T-cell responses were measured against peptide epitopes derived from ancestral SARS-CoV-2 virus and the omicron variant. Marginal increase in B and T-cell responses was observed post-dose 4.
This study provides valuable and immediate information in the context of the COVID-19 pandemic’s current stage and of vaccination programs, albeit the study is quite small (only 5 vaccinees with 4th dose), in vitro, provides limited information on functional immune responses (no antibody neutralizing activity measurements), and therefore it seems pragmatic to temper conclusions about the potential benefit or lack thereof from additional vaccine doses, based solely on this study.
For the authors’ consideration, I suggest the following modifications, that I believe will enhance the manuscript:
Line 28: revise “advantage” to “potential benefit”
Line 30: revise “effective” to “effectiveness”
Line 35: revise “highly beneficial” to “clearly beneficial”
Line 38: revise “additive” to “incremental”
Line 40: revise “benefit” to “potential benefit”
Line 43: insert number of study participants, as in “Participants ≥60 years (n=16)….”
Line 96: delete speculative assertion, “Therefore, it is expected that T cell response be maintained against VOC.” Or revise sentence citing evidence with references to support this inference.
Line 98-100: revise overreaching conclusionary sentence, “Thus, given the low decay kinetics of memory B and T cells [8] our findings question the benefit of an early boost.” This is a small study of 16 participants with in vitro measurements, and it is a leap to make comparisons with benefits that are typically defined from much larger studies that measure clinical efficacy and real-world vaccine effectiveness. I suggest making a more measured statement, perhaps referring to:
- Bar-On et al. (2022). "Protection by a Fourth Dose of BNT162b2 against Omicron in Israel." N Engl J Med. (ref 6); and
- Regev-Yochay, G., et al. (2022). "Efficacy of a Fourth Dose of Covid-19 mRNA Vaccine against Omicron." N Engl J Med 386(14): 1377-1380
